# Increased Expression of Interferon-Induced Transmembrane 3 (IFITM3) in Stroke and Other Inflammatory Conditions in the Brain

**DOI:** 10.3390/ijms23168885

**Published:** 2022-08-10

**Authors:** Elisabeth Harmon, Andrea Doan, Jesus Bautista-Garrido, Joo Eun Jung, Sean P. Marrelli, Gab Seok Kim

**Affiliations:** Department of Neurology, McGovern Medical School, The University of Texas Health Science Center at Houston, Houston, TX 77030, USA

**Keywords:** stroke, gliosis, IFITM3, microglia, interferon, aging, pdMCAO

## Abstract

Microglia, the resident innate immune cells of the brain, become more highly reactive with aging and diseased conditions. In collaboration with other cell types in brains, microglia can contribute both to worsened outcome following stroke or other neurodegenerative diseases and to the recovery process by changing their phenotype toward reparative microglia. Recently, IFITM3 (a member of the “interferon-inducible transmembrane” family) has been revealed as a molecular mediator between amyloid pathology and neuroinflammation. Expression of IFITM3 in glial cells, especially microglia following stroke, is not well described. Here, we present evidence that ischemic stroke causes an increase in IFITM3 expression along with increased microglial activation marker genes in aged brains. To further validate the induction of IFITM3 in post-stroke brains, primary microglia and microglial-like cells were exposed to a variety of inflammatory conditions, which significantly induced IFITM3 as well as other inflammatory markers. These findings suggest the critical role of IFITM3 in inducing inflammation. Our findings on the expression of IFITM3 in microglia and in aged brains following stroke could establish the basic foundations for the role of IFITM3 in a variety of neurodegenerative diseases, particularly those that are prevalent or enhanced in the aged brain.

## 1. Introduction

Microglia, astrocytes, and oligodendrocytes account for 50% of cells in the brain. These glial cells are known to support neuronal function and maintain homeostasis in the brain [1,2]. When the brain is challenged by various damages, infections, and other stressors, these glial cells, especially resident microglia, need to respond to these situations [3]. They respond by cytokine/chemokine production and secretion, proliferation, and migration to the injured site, as well as by modulating the capability of phagocytosis [4,5,6]. In aging brains, microglia undergo deleterious changes, which contribute to the initiation of a pro-inflammatory response causing neuronal loss via neurotoxic effects and facilitate pathophysiological aging processes in the aged brain [7,8]. To develop effective methods of modulating microglial functions, we need to understand the details of the transcriptional changes taking place in microglia and identify the genes that are critically involved in this detrimental process in various inflammatory situations.

Interferon signaling is known to initiate endogenous defensive programs by rapidly expressing interferons (IFNs) and interferon related genes in response to the invasion of pathogens, bacterial materials, and viruses [9]. Besides the rapid expression of interferons, interferon related genes, and roles for host defense against viral infection, emerging lines of evidence suggest that IFN signaling, mediated by Type I and Type II interferons, is critically involved in both innate and adaptive immunity (e.g., immune cell activation, proliferation and differentiation) [10]. The initial increase of interferons, mainly Type I IFN (IFNα and IFNβ) and Type II IFN (IFNγ), further amplify the signals by binding to their cognate receptors (e.g., IFNAR1, IFNAR2, IFNGR1, and IFNGR2) expressed on neighboring cells [11]. This receptor/ligand binding turns on the expression of so-called interferon-stimulated genes (Isg) [12,13]. Interestingly, emerging lines of evidence demonstrated using various neurodegenerative disease animal models also suggest detrimental interferon signaling, either Type I or Type II, contributes to neurodegeneration. These findings suggest a potential significant, but underappreciated, role of interferon signaling in initiating neuroinflammation and further neurodegeneration [14,15,16].

Interestingly, recent findings suggest that Interferon-induced transmembrane protein 3 (IFITM3), a gene that is induced by interferon signaling, is induced in astrocytes in AD brains and acts as a potent gamma-secretase activator leading to increased production of amyloid beta peptides [17]. This link of IFITM3 with Alzheimer’s disease (AD) pathology, an aging-related neurodegenerative disease, led us to study the expression of IFITM3 and other IFITM family genes and identify the role of IFITM3 in stroke in the aged brain. In AD, increased expression of IFITM3 is found in astrocytes and neurons where amyloid-beta levels are increased, however, the expression of IFITM3 in stressed microglia in various neurodegenerative diseases such as ischemic stroke has not been examined. Thus, we sought to explore the expression of IFITM3 in microglia in response to a range of inflammatory situations in vitro and in the aged brain following ischemic stroke.

The expression patterns and role of genes belonging to the IFITM family, especially IFITM3, in aged brains following stroke or with stressed primary microglia has not been revealed. In this study, we present data demonstrating that IFITM3 is a critically induced gene in various inflammatory situations, including in post-stroke brains in vivo and pro-inflammatory cytokine-stimulated primary microglia and microglial cell lines in vitro. This fundamental study will provide the solid foundation for future exploration of the role of IFITM3 in aging-related neurodegenerative diseases, including stroke, and to further identify IFITM3 as a critical mediator between neuroinflammation and neurodegeneration in the brain.

## 2. Results

### 2.1. Ischemic Stroke Increases the Expression of IFITM3 Protein in the Striatum and Peri-Infarct Cortex in Aged Brains Following Stroke

Ischemic stroke includes a significant inflammatory component to the injury. Therefore, we sought to determine if and where ischemic stroke induces expression of IFITM3 protein in the brain. The risk of ischemic stroke is greatly increased with aging, therefore, to increase the translational value of these studies, we utilized aged mice (18–22 months). Brains were isolated at post-stroke day (PSD) 7 and 14 for immunostaining for IFITM3. Stitched images show increased expression of IFITM3 in the striatum and peri-infarcted cortex 7 days after stroke and this induction was reduced at 14 days after stroke (Figure 1A). At PSD 7, IFITM3 expression was increased in ipsilateral striatum compared to naïve brain or contralateral striatum (*n* = 3–5, * *p* < 0.05, ** *p* < 0.01). At PSD 14, expression of IFITM3 in ipsilateral striatum was reduced compared with PSD 7. Expression showed a near-significant increase compared with naïve striatum (*p* = 0.053) but not contralateral striatum (*p* = 0.22) (Figure 1B). These data reveal that ischemic stroke induces profound, early expression of IFITM3 in the striatum of damaged brains that declines by two weeks.

Next, we examined the expression of IFITM3 in different peri-infarct cortex regions of stroked brains at PSD 7 and 14. We quantified the IFITM3 stained area in three different regions of interest (ROI) within the peri-infarcted area (ROIs shown in Figure 2A). Our results revealed a significant upregulation of IFITM3 protein in ROI #1 and #3 at both PSD 7 and 14 compared to sham (Figure 2B,C, *n* = 3–4, *p* < 0.05 sham vs. PSD 7 or PSD 14, one-way ANOVA with Tukey’s post hoc tests). In ROI #2, the apparent increase in IFITM3 expression did not reach statistical significance (Figure 2D, *n* = 3–4, *p* = 0.369 sham vs. PDS 7, *p* = 0.2 sham vs. PSD 14). These data demonstrate upregulation of IFITM3 protein varies by the regions in the peri-infarct cortex through PSD 14.

### 2.2. Ischemic Injury Causes the Induction of IFITM3 in Aged Brains Following Stroke

We next examined the expression profile of IFITM family members (*Ifitm1*, *Ifitm2*, and *Ifitm3*) in the post-stroke brain of aged mice. Brains were harvested at PSD 3 and 14 for quantitative mRNA analysis (qRT-PCR). Because we and others have found significant microgliosis in the thalamus at PSD 14 (secondary injury) [18], we dissected brains into cortex and thalamus to identify possible regional differences in gene expression and to explore the role of IFITM3 in the mechanism of spreading secondary injury (Figure 3A). We measured the expression of *Cst7* as a surrogate for ischemic injury induced-microglial activation and neuroinflammation. *Cst7* was increased in the cortex and thalamus at PSD 3 and 14, confirming the activation of MG in these dissected brain regions (data not shown). The expression of *Ifitm1* and *Ifitm2* were not significantly altered in the cortex or thalamus at PSD 3 or 14 (Figure 3B–E). Expression of *Ifitm3* was significantly increased in the cortex at PSD 3 (*p* < 0.05 vs. sham) and appeared to decline by PSD 14 (*p* = 0.08 vs. sham) (Figure 3F). Expression of *Ifitm3* was not altered in the thalamus at PSD 3 or 14 (Figure 3G). These data show an early transient increase in expression of *Ifitm3* in the post-stroke cortex, but not thalamus.

### 2.3. Lipopolysaccharide (LPS) Treatment Increases Ifitm3 mRNA in Primary Microglia

We found that *Ifitm3* mRNA was upregulated in ischemic brains along with the microglial activation marker gene, *Cst7*, prompting speculation that microglial activation contributes to the expression of *Ifitm3*. To test whether inflammatory stress can cause the induction of *Ifitm3* in primary microglia, we treated cells with 10 ng/mL of LPS for 6 h and then performed qRT-PCR to measure gene expression. To confirm that our LPS treatment could initiate inflammation in microglia, we first checked the level of *Il1b*, a typical marker gene for microglial activation and neuroinflammation. Indeed, we found a significant induction of *Il1b* mRNA in LPS-stimulated microglia, compared to control-treated microglia (Figure 4A). LPS-treated microglia also showed significantly increased expression of *Ifitm3* (Figure 4B, *n* = 4 or 5, ** *p* < 0.01); however, Ifitm2 was not changed by LPS treatment (Figure 4C). Correlation analysis revealed that expression of *Ifitm3* was positively correlated with the expression of *Il1b* (Figure 4D, r = 0.908), implicating *Ifitm3* induction in the microglial inflammatory phenotype.

### 2.4. Treatment with Pro-Inflammatory Cytokines Induces Ifitm3 mRNA in Primary Microglia and Sim-A9 Cells

We found a significant induction of *Ifitm3* (but not *Ifitm2*) in LPS-treated microglia. Next, we asked if in vitro exposure to pro-inflammatory cytokines (TNFα and IFNγ) can induce IFITM3 in microglia. These cytokines were chosen based on their role in inflammation in the stroke brain. Primary microglia were exposed to TNFα (20 ng/mL) and IFNγ (20 ng/mL) for 6 h and *Ifitm3* and *Ifitm2* mRNA expression was measured by qRT-PCR. Similar to LPS treatment, *Ifitm3* mRNA was significantly increased in TNFα/IFNγ treated microglia, compared to control treated microglia (Figure 5A, *n* = 5 or 6, * *p* < 0.05). However, in contrast to the insignificant Ifitm2 level changes with LPS treatment, co-treatment with pro-inflammatory cytokines induced a mild increase in *Ifitm2* mRNA (Figure 5B, *n* = 6 or 7, * *p* < 0.05).

We also checked gene expression at 24 h after treatment with TNFα and IFNγ in primary microglia. As expected, the *Il1b* level was significantly up-regulated by co-treatment of TNFα and IFNγ after 24 h, confirming microglial activation (Figure 6A, *n* = 7–8, ** *p* < 0.01). We also found profound induction of *Casp1*, a key player in the NLRP3 inflammasome which is responsible for spreading inflammatory signals to neighboring cells in response to infections by foreign pathogens and inflammatory stress (Figure 6B). TNFα and IFNγ also promoted significant expression of *Ifitm3* in primary microglia (Figure 6C, *n* = 7–8, ** *p* < 0.01). Interestingly, *Ifitm3* induction was well correlated with *Casp1* expression in microglia (Figure 6D), suggesting possible involvement of IFITM3 in inflammasome activation or response in microglia.

To determine if TNFα and IFNγ treatment can induce IFITM3 protein in primary microglia, we treated primary microglia with TNFα/IFNγ and performed western blot analysis. As shown in Figure 7, 24 h exposure to TNFα/IFNγ resulted in significant induction of IFITM3 protein in primary microglia (*n* = 6, *p* < 0.01).

We also utilized Sim-A9 cells, an immortalized murine microglial cell line. This murine microglial-like cell line is known to respond quickly to inflammatory stress and exhibit phagocytic characteristics which are common features of CNS resident microglia. We tested whether TNFα and IFNγ treatment on Sim-A9 cells could induce *Ifitm3* mRNA expression. Interestingly, *Ifitm3* expression was induced to a greater degree by treatment with 20 and 40 ng/mL of TNFα and IFNγ in Sim-A9 cells compared with primary microglia (Figure 8, *n* = 4–8, * *p* < 0.05 compared to control).

### 2.5. Amyloid-Beta Peptide Increases the Expression of Ifitm3

It is well known that sustained inflammation can facilitate amyloid pathology. In addition, amyloid beta production from APP cleavage can further reinforce the inflammatory situation in AD brains. Since we found a significant induction of IFITM3 in response to various inflammatory stressors (LPS, TNFα, and IFNγ) and recent findings suggest a novel role of IFITM3 in amyloid pathology, we sought to examine whether amyloid peptide could induce IFITM3 expression in primary microglia. We used soluble amyloid beta peptide 25–35 (Aβ25–35), which is known to show greater neurotoxicity to neurons compared to Aβ40 and Aβ42 and to activate the microglia in vitro and in vivo. We treated primary microglia with Aβ25–35 for 24 h at different concentrations (0, 10, and 40 µM). Aβ25–35 treatment (10 and 40 µM) induced increased expression of *Il-1b*, an inflammatory marker gene (Appendix A). Interestingly, expression of *Ifitm3* was not increased by 10 µM Aβ25–35 at 24 h, but was increased by 40 µM Aβ25–35 (Appendix A). These data suggest that amyloid peptide may have the capability to induce *Ifitm3* via non-pro-inflammatory receptor mediated signaling in microglia.

## 3. Discussion

In this study, we demonstrated that among IFITM family members, IFITM3 expression is selectively induced in microglia exposed to inflammatory conditions in vitro and in vivo. We showed increased IFITM3 expression in primary microglia and Sim-A9 cells exposed to pro-inflammatory mediators and found increased IFITM3 expression in peri-infarct regions of aged brains following stroke. Expression of IFITM3 was most prominent in the peri-infarct cortex, but was also increased in the striatum below the infarct. The functional role of IFITM3 is not yet well defined. However, a member of the IFITM gene family, IFITM3, was recently identified as an essential component in AD brains that links neuroinflammation and amyloid pathology, acting as a gamma secretase activator [17]. Since AD is another aging-related disorder with upregulation of IFITM3, we speculated that induction of IFITM3 may also play a role in stroke brains. In particular, we hypothesized that IFITM3 plays a role in the spreading of inflammation in the peri-infarcted region and other brain regions associated with the distal MCAO model pathology. Given the role of inflammation in enhancing amyloid-dependent pathologies (e.g., AD), we also examined if Aβ peptides could increase *Ifitm3* expression. Interestingly, exposure to Aβ25–35 promoted increased *Ifitm3* expression, suggesting a potential role of IFITM3 induction in the enhanced production of Aβ peptides. Together, these findings provide a novel foundation for the role of microglial *Ifitm3* expression in response to inflammatory stress and amyloidosis in the brain. Future studies will be required to further define the role of IFITM3 as an initiating factor versus a functional player in the inflammatory phenotype of microglia.

Originally identified as downstream signal molecules induced by interferons (IFN I and IFN II), the functions of the IFITM family were shown to act in anti-virus host defense mechanisms against a broad range of RNA viruses, including Influenza [19,20]. Acutely expressed and located in endosomal membranes and plasma membranes, these IFITM family members can counteract the virus invasion via different mechanisms (inhibiting virus entry or facilitating degradation of virus in endosome and lysosomes) [21,22,23]. The IFITM family (in humans, five IFITM members: IFITM1, IFITM2, IFITM3, IFITM5, and IFITM10; in mice, seven Ifitm members: IFITM1, IFITM2, IFITM3, IFITM5, IFITM6, IFITM7, IFITM10) was first discovered as interferon-induced genes in human neuroblastoma cells [24,25,26]. In promoter analysis, multiple putative IFN-stimulated factor (such as IRFs) binding sites were identified, suggesting their expression could depend on the degree of Type I and Type II IFN signaling [27,28]. The existence of multiple IFITM family genes on an individual chromosome (in mice, chromosome 7) might be useful to combat the spectrum of viral entry mechanisms by expressing different types and levels of IFITM at the same time upon virus invasion [29]. Yet, in the situation of brain inflammation, as we detected here, the degree of expression among IFITM family members was quite different.

Interestingly, in microglia stressed by pro-inflammatory cytokines, expression levels of IFITM1, IFITM2, and IFITM3 were quite different in response to inflammatory stress, suggesting that, even though these genes are located on the same chromosome, in addition to the molecular mechanism of expression of the IFITM family as defensive tools to prevent viral entry, other IFITM regulatory mechanisms (different induction mechanism of the expression in viral host defense) to induce IFITM family members exist, such as signaling pathways that can lead to activation of specific transcription factors which could bind to the promoter regions of IFITM family genes.

Emerging evidence suggests that IFITM3 plays a critical role in immune cells such as T cells for T-helper cell differentiation and in limiting inflammation, suggesting the active engagement of IFITM3 in adaptive immunity [26]. Interestingly, IFITM3 has been found in late endosome or lysosome where the lysosomal marker, Lamp1, was dominantly expressed in virus-invaded cells [21,30]. These findings could reflect the role of IFITM3 in brain cells, suggesting the role of IFITM3 in modulating microglial function (as well as the functions of infiltrated monocyte or macrophages) in brains, especially microglia phagocytosis, which could be detrimental or beneficial, depending on the stage of stroke. In our study, we detected increased levels of IFITM3 in inflammatory conditions in the brains of aged mice (e.g., ischemic stroke). Our in vitro findings (increased expression of IFITM3 in activated microglia) suggest a possible, novel role of IFITM3 in microglia. Increased IFITM3 may modulate a microglial phenotypical shift from the resting state to that of slightly activated microglia to more pro-inflammatory or reparative microglia. Further study is warranted to determine if IFITM3 has the capacity to modulate microglial phenotype versus acting as a component or amplifier of the inflammatory phenotype.

We speculate that IFITM3 can act as a key protein in the plasma membrane for spreading inflammation to neighboring microglia in diseased brains. This potential would be further supported by the role of IFITM3 in gamma secretase activation, which leads to increased Aβ production and inflammation in AD [17]. In another study with T-cells, IFITM3/Ifitm2 were involved in exacerbating allergic airway inflammation, suggesting the critical role of IFITM3 in spreading inflammation [31]. Future studies are needed to evaluate the role of IFITM3 in modifying neuroinflammation via infiltrating peripheral immune cells in stroke, AD, and other neurodegenerative diseases.

Our in vitro data cannot fully reflect the situation where chronic inflammation prevails, such as in AD or other neurodegenerative disease. Thus, we wanted to demonstrate increased IFITM3 expression in a diseased animal model, where neuroinflammation is the driving factor of neurodegeneration. In order to check if chronic inflammation could induce IFITM3 in inflamed brain tissue, we have chosen the pdMCAO model (primary infarction in the cortex at an early time point and secondary gliosis in the thalamus at a late time point, which are frequently detected in the brains of human stroke patients). We analyzed the expression at 2 weeks following stroke. We have found that IFITM3 levels were acutely increased and this increase was sustained through day 14 (compared to sham) in the cortex, indicating that increased expression levels of IFITM3 can be maintained via acute inflammatory stress as well as chronic inflammatory environments. These data also suggest that there are different inflammatory signaling pathways in microglia for induction of IFITM3 acutely and chronically in different regions of the brain.

One limitation of our study is the use of primary microglia isolated from neonatal (P1–2) mouse brains instead of from aged, 18–20-month-old, mice. Even though microglia can be obtained from mixed glial cell cultures (astrocytic bed) from the brains of 18–20-month-old mice, microglia that are generated on the astrocytic bed are not the same as activated, resident microglia in aged brains, in terms of transcriptional status, active gene expression, and biological age (predicted by the length of telomere, amount of lipofuscin, β-gal stained particles, etc.). Therefore, as an alternative approach to demonstrate the effect of aging in microglia on IFITM3-mediated responses and IFITM3 expression, future studies could use ex vivo purified microglia from the brains of 18–20-month-old mice. In the absence of the astrocyte bed, we would need to supplement the microglia with colony-stimulating factor 1 (CSF1) to enhance the survival of purified microglia. Another limitation of our studies is that all experiments were performed with male mice or cultured cells of mixed sex. Therefore, these studies cannot evaluate potential sex-specific responsiveness of IFITM3 expression. Future studies will need to be designed in age-matched mice of both sexes (young and aged) and in sex-specific cell culture.

In summary, we demonstrated the profound induction of IFITM3 in microglia exposed to inflammatory situations with in vitro microglial cells and in vivo stroked brains in aged mice. These studies identify IFITM3 as a new player involved in microglial phenotype modulation in response to a variety of inflammatory conditions. Future studies will be required to clarify whether IFITM3 has the capacity to modulate microglial phenotype versus whether it is induced as a result of microglial phenotypical change in conditions of neurodegenerative diseases.

## 4. Materials and Methods

### 4.1. Animals

All procedures were performed in accordance with NIH guidelines for the care and use of laboratory animals and were approved by the Institutional Animal Care and Use Committee of the University of Texas Health Science Center. In vivo studies were carried out in compliance with the ARRIVE guidelines [32]. Male C57BL/6J mice (18–22 months old: aged) were used to examine gene expression changes following stroke.

### 4.2. Permanent Distal Middle Cerebral Artery Occlusion (PDMCAO) Model

We used male C57BL/6 mice aged 18–22 months. Permanent focal cerebral ischemia was induced by cauterizing the right distal middle cerebral artery (MCA) using electro-coagulation (Accu-temp variable low temperature cautery) [18]. For the surgical procedure, the mice were briefly anesthetized with isoflurane (4% induction and 2% maintenance in airflow). The body temperature (rectal probe) was maintained at 37 °C with a feedback-controlled heating pad. The temporal muscle was detached from the skull using forceps and the MCA was identified through the semi-transparent skull. Using a micro-drill, a burr hole was generated just above the middle cerebral artery. The low temperature cautery was used to permanently ligate the MCA. During recovery, the mice were put in a custom-built warming chamber [33] for two hours to maintain normothermic body temperature. Sham controls were generated by following the same procedure minus the electro-coagulation of the MCA.

### 4.3. Brain Sample Preparation and Immunostaining

Brains were harvested and then fixed for 24 h in 4% PFA at 4 °C. The brains were then transferred to 30% sucrose solution in PBS for an additional 24 h prior to generating 30 µm coronal sections with a frozen microtome (Micron HM 450, Thermo Fisher Scientific, Waltham, MA, USA). To begin immunostaining, sections were washed with PBS, incubated with blocking buffer (10% goat serum, 0.3% Trion X-100 in PBS), and incubated overnight at 4 °C with the following primary antibody: Rabbit anti-IFITM3 antibody (Thermo Fisher Scientific, Waltham, MA, USA). We used a donkey anti-rabbit IgG-Alexa 594 (1:200, Thermo Fisher Scientific, Waltham, MA, USA) as the secondary antibody. Sections were incubated with DAPI (4′,6-diamidino-2-phenylindole) to label nuclei. Images were obtained using a Leica TCS SPE confocal system and a Leica DMi8 fluorescence microscope system (Leica Biosystem, Richmond, IL, USA). Multiple images of the whole brain or individual hemispheres were captured with the 10× objective and stitched to generate a single high-resolution image (Leica LAS X software ver. 3.6.0.20104, Leica Biosystem, Nussloch, German). Image analysis was performed using ImageJ software (ver. 1.53q, National Institutes of Health, Bethesda, MD, USA).

### 4.4. Primary Microglia Culture

To isolate primary microglia, a mixed glial cell culture was first established from P1 postnatal brains of C57BL/6 mice. Cortices were obtained from the postnatal brains and were dissociated by trypsin/EDTA incubation for 15 min at 37 °C. Cells were then collected, re-suspended in DMEM by multiple pipetting, and passed through a 75 µm cell strainer. The cell suspension was seeded into T75 tissue culture flasks (one mouse pup brain per T75 flask) and maintained in Dulbecco’s Modified Eagle’s Medium (Thermo Fisher Scientific, Waltham, MA, USA) supplemented with 20% fetal bovine serum (FBS; HyClone, Logan, UT, USA) and 1% penicillin/streptomycin and grown as a mixed glial culture for 10 to 14 days. Once the mixed glial cultures were completely confluent (i.e., microglia grown on the monolayer of the astrocytic bed with a bright, round shape), the flasks were shaken at 200 rpm for two hours at 37 °C to detach microglia from the astrocytic bed. Detached microglia were pelleted at 1000× *g* for three minutes at 4 °C. Primary microglia were seeded on six-well culture plates (2 × 10^5^ cells/well) for RNA isolation and qRT-PCR.

### 4.5. Sim-A9 Cell Culture

The immortalized Sim-A9 microglial cell line was derived from brains of C57BL/6 neonate mice and is known to possess many functional properties of primary microglia [34]. Sim-A9 cells were cultured at 37 °C, 5% CO_2_ in Dulbecco’s modified Eagle’s medium (DMEM; Gibco, Life Technologies Corporation, Grand Island, NY, USA), supplemented with 10% heat-inactivated fetal bovine serum (FBS; HyClone, Logan, UT, USA) and 5% horse serum (HS; HyCLone, Logan, UT, USA). Sim-A9 cells were seeded on six-well culture plates (1 × 10^5^ cells/well) for RNA isolation and qRT-PCR and grown to 70–80% confluency before treatment with pro-inflammatory cytokines, LPS, and amyloid β.

### 4.6. RNA Isolation and qRT-PCR

Total RNA was isolated from the cortex and thalamus of sham and post-stroke brains at post-stroke day (PSD) 7 and 14 using TRIzol Reagent (Thermo Fisher Scientific, Waltham, MA, USA) and RNeasy Mini Kit (Qiagen, Hilden, Germany). For RNA isolation from primary cells and Sim-A9 cells, we used only the RNeasy Mini Kit and followed the instructions provided by the manufacturer. After isolation, RNA concentration and purity were measured using a NanoDrop spectrophotometer (Thermo Fisher Scientific, Waltham, MA, USA). The 260/280 of all RNA samples collected in TE buffer were >1.8. Synthesis of cDNA was performed using iScript™ Reverse Transcription Supermix (Bio-Rad, Hercules, CA, USA). Next, mRNA expression was analyzed using SsoAdvanced™ universal SYBR^®^ Green Supermix (Bio-Rad, Hercules, CA, USA) on a CFX384 Touch real-time PCR cycler (Bio-Rad, Hercules, CA, USA). *Gapdh* was used as an internal control. The relative expression of the target gene was normalized to *Gapdh* expression using the 2 ΔΔCt analysis method. The primer sequences were as follows (Table 1).

### 4.7. Western Blotting

Cell lysates were obtained using the RIPA lysis buffer containing cOmplete™ Protease Inhibitor Cocktail (MilliporeSigma, Burlington, MA, USA) from primary microglia treated with TNFα and IFNγ for 24 h. Protein concentrations were measured using the BCA method (Pierce BCA assay kit, Thermo Fisher Scientific, Waltham, MA, USA) with BSA standard. Twenty micrograms of protein from each sample was loaded equally into the wells of Mini-PROTEAN TGX™ 4–15% protein gels (Bio-Rad, Hercules, CA, USA). After running SDS-PAGE, polyvinylidene difluoride (PVDF) membranes were used to transfer protein from gels to membranes. Membranes were blocked with 3% skim milk in TBS/Tween 0.3% for 30 min. Primary antibody against mouse IFITM3 (Catalog # PA5-11274, Thermo Fisher Scientific, Waltham, MA, USA) was diluted at 1:1000 in 3% skim milk in TBS/T and membranes were incubated in diluted IFITM3 antibody for 24 h at 4 °C. β-Actin antibody (Cat# A2228, MilliporeSigma, Burlington, MA, USA) was used to evaluate protein loading and to normalize IFITM3 intensity. Pierce™ ECL Western Blotting Substrate (Thermo Fisher Scientific, Waltham, MA, USA) was used to visualize the bands on the membranes.

### 4.8. Statistical Analysis

The Prism 7.03 program (GraphPad Software, San Diego, CA, USA) was used to create graphs and perform statistical analysis. The data are expressed as the mean ± standard deviation (SD). An unpaired Student’s *t*-test was used for comparisons between two groups and the differences among multiple groups were determined by one-way ANOVA with multiple comparisons test. Correlation was determined by Pearson correlation analysis.

## Figures and Tables

**Figure 1 ijms-23-08885-f001:**
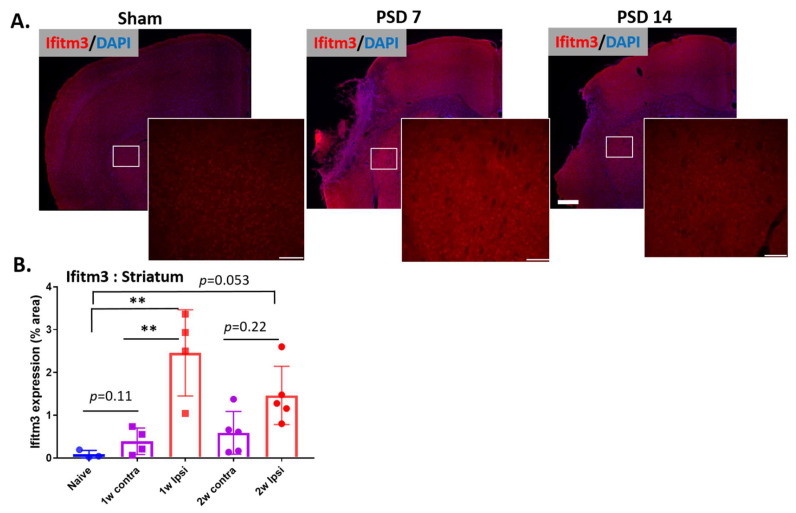
IFITM3 expression was increased in ipsilateral striatum. To check in vivo protein expression of IFITM3, PDMCAO was performed with aged mice (18–20 months, male). At 1 week and 2 weeks after stroke, brains were isolated and IFITM3 immunostaining was performed with the brain sections. (**A**) Stitched and 20× images show increased intensities of IFITM3 in the striatum at PSD 7 and PSD 14 compared to aged naïve and contralateral striatum (*n* = 3–4, scale bar = 0.5 mm, 100 µm; insert). (**B**) Quantification revealed increased expression of IFITM3 in the ipsilateral striatum compared to contralateral striatum and naïve striatum (*n* = 3–5, ** *p* < 0.01, one way ANOVA followed by Tukey’s multiple comparison test).

**Figure 2 ijms-23-08885-f002:**
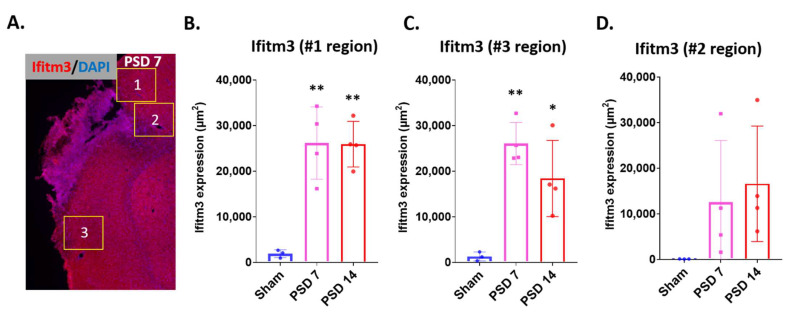
Increased expression of IFITM3 was found in the peri-infarct cortex. At 1 week and 2 weeks after stroke, brains were isolated and IFITM3 immunostaining was performed with the brain sections. (**A**) Quantification was performed with the 20X images taken from 3 different areas of peri-infarcted cortex. The yellow rectangles indicate the areas where images were taken. (**B,C**). Quantification shows that IFITM3 was significantly increased in areas #1 and #3 of the peri-infarcted cortex, compared to sham (*n* = 3–4, * *p* < 0.05, ** *p* < 0.01 sham vs. PSD 7 or PSD 14, one-way ANOVA with Tukey’s post hoc tests). (**D**). Area #2 of the peri-infarct cortex showed a non-significant trend for increased IFITM3 expression.

**Figure 3 ijms-23-08885-f003:**
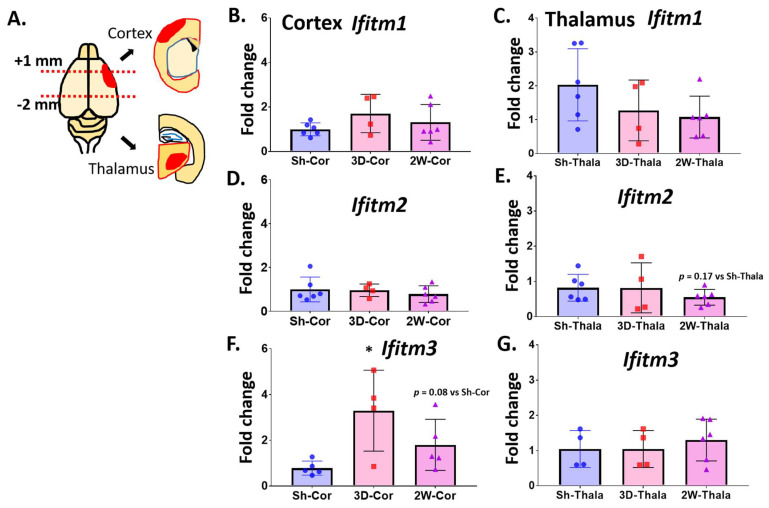
*Ifitm3* is markedly induced in the damaged cortex of post-stroke brains. Ischemic stroke was induced by permanent distal middle cerebral artery occlusion using C57BL6 male mice (18–20 months old). At 3 days and 2 weeks after surgeries, brains were separated into cortex and thalamus (**A**). After isolation of RNA, qRT-PCR was performed to measure gene expression for IFITM family genes. *Ifitm1* levels in (**B**) the cortex and (**C**) thalamus of sham, PSD 3 and PDS 14 brains was measured by qRT-PCR (*n* = 4–6). *Ifitm2* levels in (**D**) the cortex and (**E**) thalamus of sham, PSD 3 and PDS 14 brains was measured by qRT-PCR (*n* = 4–6, *p* = 0.17 vs. Sh-Thala). *Ifitm3* levels in (**F**) the cortex and (**G**) thalamus of sham, PSD 3 and PDS 14 brains was measured by qRT-PCR (*n* = 4–6, * *p* < 0.05). One-way ANOVA with Tukey’s post hoc tests, *p* = 0.17 vs. Sh-Thala, *p* = 0.08 vs. Sh-Cor, two-tailed unpaired Student’s *t*-test.

**Figure 4 ijms-23-08885-f004:**
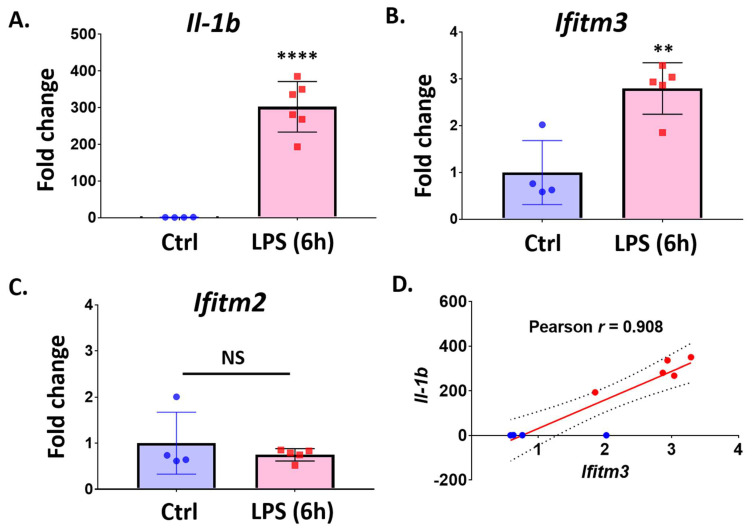
*Ifitm3* expression is induced by LPS treatment in primary microglia. Primary microglia were treated with LPS (10 ng/mL) for 6 h. qRT-PCR was performed to measure gene expression for *Ifitm3* and other pro-inflammatory cytokines genes. (**A**) *Il1b* level was significantly increased by LPS treatment (*n* = 4–6, **** *p* < 0.0001 compared to control, two-tailed unpaired Student’s *t*-test). (**B**) *Ifitm3* level was significantly increased by LPS treatment (*n* = 4–5, ** *p* < 0.01 compared to control, two-tailed unpaired Student’s *t*-test). (**C**) *Ifitm2* level was not significantly changed by LPS treatment (*n* = 4–5). (**D**) Correlation analysis revealed a close correlation between *Ifitm3* and *Il1b* expression in primary MG treated with LPS (r = 0.908).

**Figure 5 ijms-23-08885-f005:**
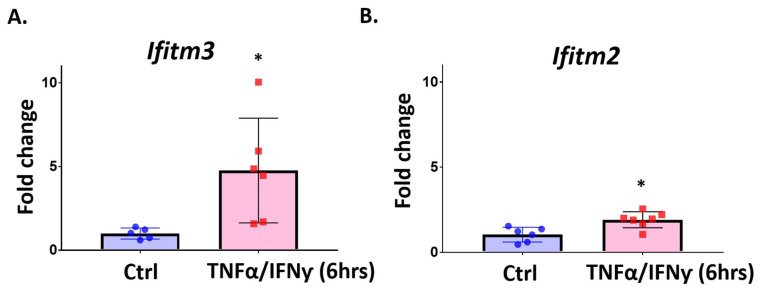
The expression of *Ifitm3* is increased at 6 h after TNFα and IFNγ treatment in primary microglia. Primary microglia were treated with TNFα and IFNγ (20 ng/mL) for 6 h. qRT-PCR was performed to measure gene expression for *Ifitm3* and *Ifitm2*. (**A**) The level of *Ifitm3* was increased by TNFα and IFNγ (*n* = 5–6, * *p* < 0.05 compared to control). (**B**) The level of *Ifitm2* was also increased by TNFα and IFNγ (*n* = 6–7, * *p* < 0.05 compared to control, two-tailed unpaired Student’s *t*-test).

**Figure 6 ijms-23-08885-f006:**
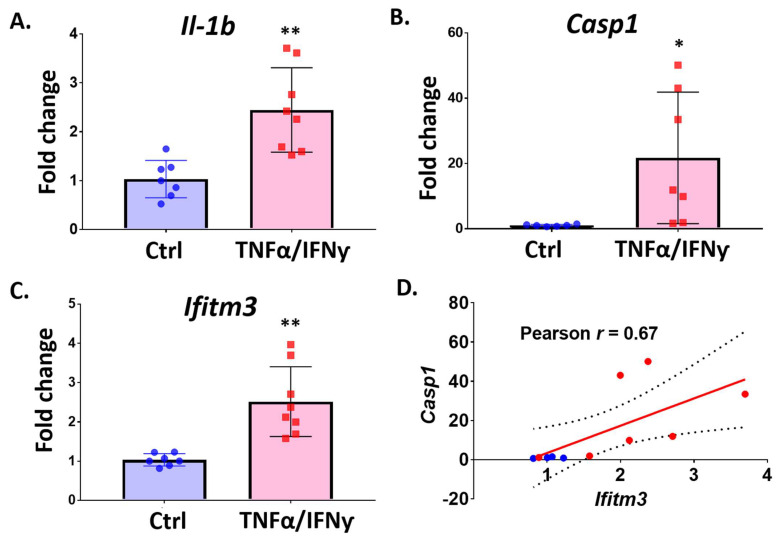
The expression of *Ifitm3* and inflammasome related genes is increased by TNFα and IFNγ in primary microglia. Primary microglia were treated with TNFα and IFNγ (20 ng/mL) for 24 h. qRT-PCR was performed to measure gene expression for *Ifitm3* and inflammasome related genes (*Casp1* and *Il1b*). (**A**) The level of *Il1b*, an inflammatory marker gene, was increased by TNFα and IFNγ (*n* = 7–8, ** *p* < 0.01 compared to control, two-tailed unpaired Student’s *t*-test). (**B**) The level of Casp1 was increased by TNFα and IFNγ (*n* = 4–8, * *p* < 0.05 compared to control, two-tailed unpaired Student’s *t*-test). (**C**) *Ifitm3* level was significantly increased by TNFα and IFNγ (*n* = 7–8, ** *p* < 0.01 compared to control, two-tailed unpaired Student’s *t*-test). (**D**) Correlation analysis showed a close correlation between *Ifitm3* and *Casp1* expression (r = 0.67, blue dots-Ctrl, red dots-TNFα/IFNγ).

**Figure 7 ijms-23-08885-f007:**
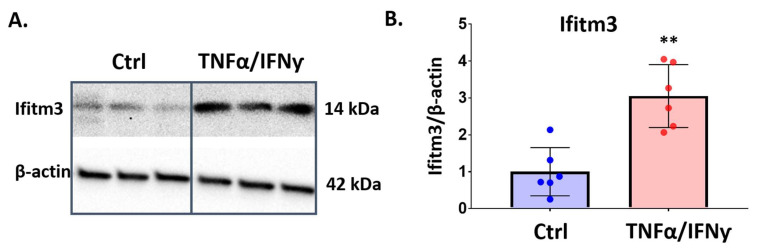
The IFITM3 protein level is increased by TNFα and IFNγ in primary microglia. Primary microglia were treated with TNFα and IFNγ (20 ng/mL) for 24 h. Western blotting was performed to measure protein level of IFITM3. (**A**) IFITM3 (seen at 14 kDa) was increased in the treatment group, compared to control. (**B**) Quantification of IFITM3 to β-actin showed the significant induction of IFITM3 protein in microglia (*n* = 6, ** *p* < 0.01, two-tailed unpaired Student’s *t*-test).

**Figure 8 ijms-23-08885-f008:**
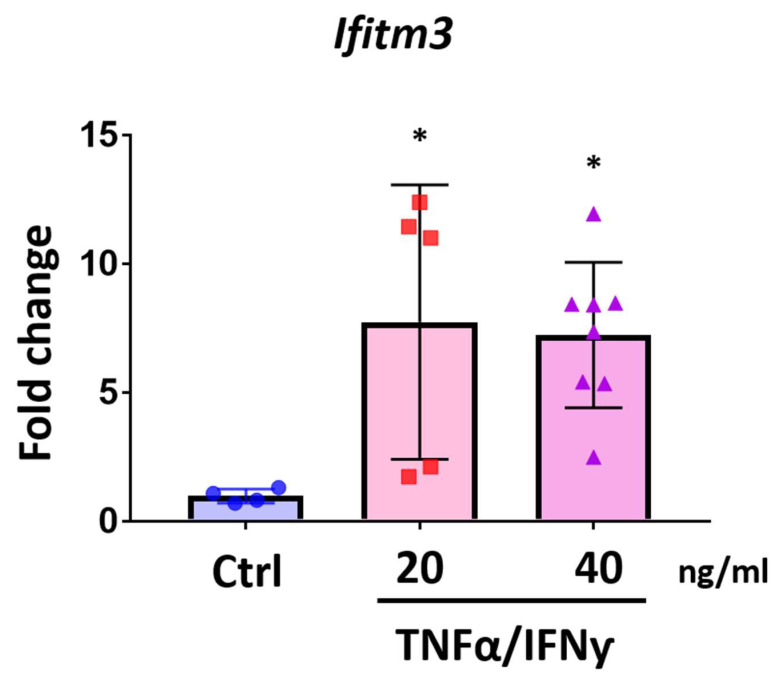
*Ifitm3* expression is increased in TNFα and IFNγ treated microglial cell line, Sim-A9 cells. Sim-A9 cells were treated with TNFα and IFNγ (20 ng/mL and 40 ng/mL) for 24 h. qRT-PCR was performed to measure gene expression for *Ifitm3*. *Ifitm3* level was significantly increased by TNFα and IFNγ (*n* = 4–8). * *p* < 0.05 compared to control, one-way ANOVA with Tukey’s post hoc tests.

**Table 1 ijms-23-08885-t001:** RT-PCR Primer Sequences (5′ to 3′).

Primers	Forward	Reverse
*Ifitm3*	TTCTGCTGCCTGGGCTTCATAG	ACCAAGGTGCTGATGTTCAGGC
*Ifitm1*	GCCACCACAATCAACATGCCTG	ACCCACCATCTTCCTGTCCCTA
*Ifitm2*	GTTCCAGAGTCAGTACCATGAG	GGCGTTGAAGAAGAGTGTATTG
*Casp1*	TCTGTATTCACGCCCTGTTG	GATAAATTGCTTCCTCTTTGCCC
*Il1b*	CTGTGTCTTTCCCGTGGACC	CAGCTCATATGGGTCCGACA
*Gapdh*	GATGGCAACAATCTCCACTTTGC	GCCGCATCTTCTTGTGCAGT

## Data Availability

All data generated or analyzed during this study are included in this published article.

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
