# Peer review of "Increased Expression of Interferon-Induced Transmembrane 3 (IFITM3) in Stroke and Other Inflammatory Conditions in the Brain"

_ijms, 2022, doi:10.3390/ijms23168885_

Round 1
Reviewer 1 Report
Ifitm3 has been described as a molecular mediator between amyloid pathology and neuroinflammation. The research manuscript titled as ‘Increased Expression of Interferon-Induced Transmembrane 3 (Ifitm3) in
Stroke and Other Inflammatory Conditions in The Brain’ by Harmon et al. investigated the expression levels of Ifitm3 in microglia under post-stroke brains, or various inflammatory conditions. However, study should have been conducted in this manuscript regarding the functions of Ifitm3 in microglia under post-stroke brains, or various inflammatory conditions, to get interests for most researchers on neuroscience and molecular medicine etc.
Author Response
We thank the reviewers for their time and valuable comments and suggestions to improve the manuscript. We have addressed the specific comments below and indicated new or modified text in blue.
[Comments and Suggestions for Authors from Revier #1]
Ifitm3 has been described as a molecular mediator between amyloid pathology and neuroinflammation. The research manuscript titled as ‘Increased Expression of Interferon-Induced Transmembrane 3 (Ifitm3) inStroke and Other Inflammatory Conditions in The Brain’ by Harmon et al. investigated the expression levels of Ifitm3 in microglia under post-stroke brains, or various inflammatory conditions. However, study should have been conducted in this manuscript regarding the functions of Ifitm3 in microglia under post-stroke brains, or various inflammatory conditions, to get interests for most researchers on neuroscience and molecular medicine etc.
[Responses]
We thank reviewer #1 for providing us with in-depth insight leading us to confidence in our next project. In our future projects, we will investigate the mode of action (MOA) and role of Ifitm3 in various CNS cells in aged stroke, AD and other neurodegenerative diseases. We agree with reviewer #1’s opinion and have launched new projects to elucidate the role of Ifitm3 in microglia as well as in astrocyte and endothelial cells utilizing siRNA-Ifitm3 (mouse and human), recombinant Ifitm3 protein, and Ifitm3 KO mice and reporter mice (JAX, #013139). We will also check the role of endothelial Ifitm3 in recruiting microglia to injured sites since our recent single cell RNA sequencing (scRNAseq) dataset indicates that Ifitm3 in microglia as well as endothelial cells is highly expressed in sham or control brains and is further increased by aging and ischemic stroke. Interestingly, Ifitm3 is among the top 10 genes highly regulated by ischemic insults in microglial clusters, showing a possible novel role in accelerating neuroinflammation which led us to study the expression pattern of Ifitm3 in microglia under inflammatory stress. In this project (an expected 2-year project), we hope to obtain clear evidence of microglial Ifitm3’s role in modifying microglial genes leading to further evolution of infarction and brain damage/neurodegeneration in aged brains following stroke, or in AD. Our solid data showing profound expression of Ifitm3 in microglia under inflammatory conditions, shown in this manuscript, creates a scientific premise that allows for these kinds of future projects. We have addressed the possible roles of Ifitm3 in microglia and other cells in the discussion section. I agree with reviewer #1 that it would be better to get even a small piece of evidence implying the role of Ifitm3 in directing microglial activation. We hope to report data showing the role of Ifitm3 in various CNS cells to our scientific community in the future. Ifitm3 (normally expressed and located in endosome) is a very important molecule for blocking viral infection, and interferon-related genes play critical roles in progressing pathophysiology of stroke and AD. Thus, we believe that, based on our solid expression studies, Ifitm3 is negatively involved in neuroprotection (Ifitm3 increase neuroinflammation/infarction) and neuro-repair (Ifitm3 can modulate reparative microglial activity), but positively involved in neuroinflammation (by activating ROS generation system, pro-inflammatory cytokine and Casp1 mediated NLRP3 inflammasome activation) in aged brains following stroke.

Reviewer 2 Report
Thank you for permitting me to review this manuscript
In this controlled experimental design the authors created an acute inlammatory situation in mice's brain by clamping down a middle cerebral artery, Ifitm3 and derivatives were significantly increased as part of a probable experimentally induced acute inflammatory cycle.
The authors cite the limitation of the study as being the use of primary microglia isolated from neonates mices but I didn't find an explanation why old mice microglia was not used , in addition please clarify the sentence line 318-319
There should be some reference of the similarity of mice brain with human brain in their inflammatory response to acute injury and chronic degenerative disease.
There is a difference between aged brain and Alzheimer disease which is a specific entity this should be adressed in the discussion section
While this experiment evidenced the increase in IFTM3 in acute inflammatory response , it does not necessary mean the same path in a chronic situation such as AD or other degenerative brain disease , and shoul be adressed in the discussion
Line 133 , It is not clear if thalamus was affected by the clamping except by the drawing do the authors have other imags confirming that thalamus was also affected ?
If thalamus was indeed affected how come the itfm3 ws not modulated in that area ? the authors may provide a suggestion in the discussion section
Figure 1 b ; Please provide supplementary leend
Author Response
Comments and Suggestions for Authors from Reviewer #2
Thank you for permitting me to review this manuscript
In this controlled experimental design the authors created an acute inlammatory situation in mice's brain by clamping down a middle cerebral artery, Ifitm3 and derivatives were significantly increased as part of a probable experimentally induced acute inflammatory cycle.
The authors cite the limitation of the study as being the use of primary microglia isolated from neonates mices but I didn't find an explanation why old mice microglia was not used , in addition please clarify the sentence line 318-319
[Responses]
We thank reviewer #2 for the supportive comments.
That is a great point! Thank you very much for raising this issue regarding the use of “aged microglia”.
Most of our in vitro studies were done using primary microglia isolated from “P2” mice, “young” microglia. The most important thing for us to examine was simply the induction of Ifitm3 (mRNA or protein) in microglia upon inflammatory insults. First, we needed to determine if the expression of Ifitm3 can be increased in “typical” or “young” (isolated from P2 mice) primary microglia under various inflammatory conditions since we did not know the effects of pro-inflammatory cytokines and amyloid-beta on Ifitm3 expression in primary microglia, regardless of the mouse’s age when the microglia was obtained. We were eager to know if Ifitm3 can be further expressed in microglia in brains with neurodegenerative disease compared to non-diseased conditions, since Ifitm3 expression has been reported mainly in virus-transfected cells. Also, recent excellent findings in AD (Hur, J. Y.; Nature. 2020) reported that Ifitm3 is expressed in astrocyte (interestingly, they did not report Ifitm3 expression in microglia) and that Ifitm3 is involved in the increase of gamma-secretase activity which is required for pathophysiological APP processing. We have been continuously culturing primary microglia and astrocyte from P2 mice. So, we wanted to know, first, if “neonatal”, “young” primary microglia isolated from astrocytic beds could generate mRNA and protein of Ifitm3 inside cells. The reason we choose to use “neonatal” microglia initially is our ability to easily culture primary microglia from P2 mice and obtain a high enough number of microglia with a homologous population and minimum activation status. We have generated very consistent data using young microglia, showing the increased expression of Ifitm3 (since young microglia show minimum activation status, much lower than aged microglia). However, as reviewer #2 suggested, we will test if inflammatory stress further increases the expression of Ifitm3 in aged microglia and compare that expression to those in young microglia. We have recently established microglial culture from aged mice (18-20 months) so, we plan to compare the expression of Ifitm3 using microglia from P2 mice and aged mice. If we find a significant difference in the degree of Ifitm3 expression, we will report our findings to IJMS. There are also limitations to the use of aged microglia. It has not been clearly demonstrated that microglia cultured from aged mice are really “aged” microglia. Most microglia in aged microglial culture are generated from pre-existing, aged microglia from the astrocytic bed. Do these “aged” microglia show the phenotypes of microglia in aged brains? Do newly generated microglia from aged microglia culture have similar transcriptome or activation-related gene expression to aged microglia found in the brains of aged mice? In most cases, responses to inflammatory stress will be similar between “P2” microglia and “aged” microglia but degree of induction of inflammation in microglia may differ.
For clarification of the sentence in line 318-319, since our data suggests the possible role of Ifitm3 in modulating the microglial phenotype, but does not directly indicate nor show the mode of action of Ifitm3, we have addressed in the following sentence that detailed future studies will be needed to verify the role of Ifitm3 in activated microglia. Thus, we have revised this sentence (line 318-319) as below.
“Our in vitro findings (increased expression of Ifitm3 in activated microglia) suggest a possible, novel role of Ifitm3 in microglia. Increased Ifitm3 may modulate a microglial phenotypical shift from the resting state to that of slightly activated microglia to more pro-inflammatory or reparative microglia.”
There should be some reference of the similarity of mice brain with human brain in their inflammatory response to acute injury and chronic degenerative disease.
There is a difference between aged brain and Alzheimer disease which is a specific entity this should be adressed in the discussion section
While this experiment evidenced the increase in IFTM3 in acute inflammatory response , it does not necessary mean the same path in a chronic situation such as AD or other degenerative brain disease , and shoul be adressed in the discussion
[Responses]
Thank you very much. I totally agree with reviewer #2’s opinion. The specific aspects/characteristics of microglial activation and function may differ in human and rodents. However, even though, special signaling pathways and genes that are required to activate microglia or to polarize microglial phenotypes to either pro-inflammatory or reparative microglia in human and mice are different, basic characteristics of microglia and microglial response under restricted conditions such as pro-inflammatory cytokines or amyloid beta treatment will be similar between humans and mice. Gliosis found in the human stroke patient was similar to the gliosis found in brains of mice that received “controlled” stroke surgery. Thalamic gliosis was also found in human stroke patients and cortical infarction is of course found in the cortex of animal models. The general mechanisms that induce inflammation and drive neuroinflammation in the brains of various neurodegenerative animal models are very similar to parallel events in the brains of human patients with various inflammatory diseases. This is a reason why we utilize the animal model to study the mechanism of neuroinflammation and disease progression caused by microgliosis and astrogliosis. Additionally, we will need to check if the level of Ifitm3 protein is going up in the brains of human autopsy samples to verify the increased expression in human diseased brains.
Based on our findings with primary microglia treated with pro-inflammatory cytokines or amyloid beta, our in vitro data would not fully reflect the situation of chronic inflammation found in AD or other neurodegenerative diseases. Thus, we wanted to explore the expression in a diseased animal model, pdMCAO at 2 weeks following stroke, to check if chronic inflammatory situations could also increase Ifitm3 expression. In this study, we have shown that Ifitm3 levels were acutely increased and this increase was sustained through day 14 (compared to sham) in the cortex, clearly indicating that the expression of Ifitm3 can be achieved via acute inflammatory stress as well as chronic inflammatory environments. We believe that our data also imply that different inflammatory signaling pathways in microglia are involved in the induction of Ifitm3 acutely and chronically. Thus, we have included the following paragraph in the discussion to address these issues.
“Our in vitro data can not fully reflect the situation where chronic inflammation prevails, such as in AD or other neurodegenerative disease. Thus, we wanted to demonstrate increased Ifitm3 expression in a diseased animal model, where neuroinflammation is the driving factor of neurodegeneration. In order to check if chronic inflammation could induce Ifitm3 in inflamed brain tissue, we have chosen the pdMCAO model (primary infarction in the cortex at an early time point and secondary gliosis in the thalamus at a late time point, which are frequently detected in the brains of human stroke patients). We analyzed the expression at 2 weeks following stroke. We have found that Ifitm3 levels were acutely increased and this increase was sustained through day 14 (compared to sham) in the cortex, indicating that increased expression levels of Ifitm3 can be maintained via acute inflammatory stress as well as chronic inflammatory environments. These data also suggest that there are different inflammatory signaling pathways in microglia for induction of Ifitm3 acutely and chronically in different regions of the brain.”
Line 133 , It is not clear if thalamus was affected by the clamping except by the drawing do the authors have other imags confirming that thalamus was also affected ?
If thalamus was indeed affected how come the itfm3 ws not modulated in that area ? the authors may provide a suggestion in the discussion section
[Responses]
That is a really good point. Thank you very much for providing us your scientific insight.
We have continuously observed microglial gliosis in the thalamus at 1 or 2 weeks post- stroke. Our model, pdMCAO, generates infarction in only the cortex and delayed gliosis in the thalamus. In addition to increased gliosis in the thalamic regions following stroke, we have found that the genes closely related to microglial activation such as Tyrobp, Ifi27l2a, Il1β and Cst7 were upregulated in the thalamic area following stroke. This indicates that the thalamus is definitely affected when there is primary injury in the cortex as a result of pdMCAO. Cst7 is a gene recently identified as a liable marker gene for microglial activation. We used Cst7 as a key marker gene for microglial activation and inflammation. As you can see, we saw a huge induction of Cst7 in the thalamic region 2 weeks following stroke (R-Fig.1, This data should be visible only to reviewers).
We have cited another paper of ours showing thalamic gliosis at 2 weeks following stroke and demonstrating that memantine (a NMDA antagonist) reduced primary infarction and secondary gliosis in the thalamus (Kim, G. S.; Sci Rep. 2021). Our data suggest that there could be different inflammatory signaling pathways in microglia for the induction of Ifitm3 acutely (in the cortex at day 3) and chronically (in the thalamus at day 14). Also, we cannot exclude the possibility that microglia exhibit a regional difference (e.g. microglia in grey matter, white matter, thalamic microglia, and disease associated microglia, etc.), as a reason for differential expression of Ifitm3 in the cortex and thalamus after stroke. Microglia may respond differently to inflammatory insults due to spatial location, source of inflammation, microenvironment and different transcriptome profiles. This may differentiate the acute induction of Ifitm3 in the cortex at 3 days following stroke from the non-significant (but still a gradually increasing) induction of Ifitm3 in the thalamus, even though the thalamus is affected by primary infarction. We included this in the discussion.
“We have found that Ifitm3 levels were acutely increased and this increase was sustained through day 14 (compared to sham) in cortex, indicating that the increased expression levels of Ifitm3 can be maintained via acute inflammatory stress as well as chronic inflammatory environments. These data also suggest that there are different inflammatory signaling pathways in microglia for induction of Ifitm3 acutely and chronically in different brain regions.”
Figure 1 b ; Please provide supplementary leend
[Responses]
Thank you very much, we rechecked the manuscript thoroughly for missing legends and believe we have provided all figure legends in the manuscript. Please find the figure/supplementary legend for figure 1b below and on page 13 of the manuscript.
“Figure 1. Ifitm3 expression was increased in ipsilateral striatum. To check in vivo protein expression of Ifitm3, PDMCAO was performed with aged mice (18-20 months, male). At 1 week and 2 weeks after stroke, brains were isolated and Ifitm3 immunostaining was performed with the brain sections. A. Stitched and 20x images show increased intensities of Ifitm3 in the striatum at PSD 7 and PSD 14 compared to aged naïve and contralateral striatum (n=3-4, scale bar=0.5 mm, 100 um; insert). B. Quantification revealed increased expression of Ifitm3 in the ipsilateral striatum compared to contralateral striatum and naïve striatum (n=3-5, * p<0.05, ** p<0.01, one way ANOVA followed by Tukey's multiple comparison test).”
Hur, J. Y.; Frost, G. R.; Wu, X.; Crump, C.; Pan, S. J.; Wong, E.; Barros, M.; Li, T.; Nie, P.; Zhai, Y.; Wang, J. C.; Tcw, J.; Guo, L.; McKenzie, A.; Ming, C.; Zhou, X.; Wang, M.; Sagi, Y.; Renton, A. E.; Esposito, B. T.; Kim, Y.; Sadleir, K. R.; Trinh, I.; Rissman, R. A.; Vassar, R.; Zhang, B.; Johnson, D. S.; Masliah, E.; Greengard, P.; Goate, A.; Li, Y. M., The innate immunity protein IFITM3 modulates gamma-secretase in Alzheimer's disease. Nature 2020, 586 (7831), 735-740.
Kim, G. S.; Stephenson, J. M.; Al Mamun, A.; Wu, T.; Goss, M. G.; Min, J. W.; Li, J.; Liu, F.; Marrelli, S. P., Determining the effect of aging, recovery time, and post-stroke memantine treatment on delayed thalamic gliosis after cortical infarct. Sci Rep 2021, 11 (1), 12613.

Reviewer 3 Report
Accept
Author Response
N/S
Round 2
Reviewer 1 Report
The current available data regarding the expression of Ifitm3 in this manuscript is solid, however its impact is rather limited due to the lack of the functional study of Ifitm3 in microglia under post-stroke brains, or various inflammatory conditions.